# Longitudinal prevalence and co-carriage of pathogens associated with nursing home acquired pneumonia in three long-term care facilities

Ryann N. Whealy [1,2], Alexander Roberts[1,2], Tara N. Furstenau[2], Skylar Timm[1,2], Sara Maltinsky[1], Sydney J. Wells[1], Kylie Drake[1], Kayla Ramirez[1], Candice Bolduc[3], Ann Ross[3], Talima Pearson[1], Viacheslav Y. Fofanov[1,2]*

**1** Pathogen and Microbiome Institute, Northern Arizona University, Flagstaff, Arizona, United States of America, **2** School of Informatics, Computing, and Cyber Systems, Northern Arizona University, Flagstaff, Arizona, United States of America, **3** Mobile Dentistry of Arizona, Mesa, Arizona, United States of America

* Viacheslav.Fofanov@nau.edu

## Abstract

Nursing home acquired pneumonia (NHAP), and its subset – aspiration-associated pneumonia, is a leading cause of morbidity and mortality among residents in long-term care facilities (LTCFs). Understanding colonization dynamics of respiratory pathogens in LTCF residents is essential for effective infection control. This study examines the longitudinal trends in prevalence, persistence, bacterial load, and co-colonization patterns of five respiratory pathogens in three LTCFs in Phoenix, Arizona. Anterior nares and oral swabs were collected every other week and tested using qPCR for *Haemophilus influenzae*, *Pseudomonas aeruginosa*, *Streptococcus pneumoniae*, *Staphylococcus aureus*, and *Chlamydia pneumoniae*. Weekly average positivity rates were 17.75% for *H. influenzae* (0% – 39.39%), 9.95% for *P. aeruginosa* (0% – 37.74%), 31.89% for *S. pneumoniae* (1.79% – 41.67%), and for 28.00% for *S. aureus* (0% – 55.36%). *C. pneumoniae* was not detected. *H. influenzae* and *S. pneumoniae* predominantly colonized the oral cavity, while *P. aeruginosa* and *S. aureus* predominantly colonized the nasal cavity. Colonization by *S. pneumoniae* and *S. aureus* was significantly more persistent than *H. influenzae* and *P. aeruginosa*, with persistence correlating with significantly higher bacterial loads. Co-colonization did occur in ~20% of positive samples but appeared to be due to random chance. This study reveals distinct colonization patterns among respiratory pathogens in LTCF residents, highlighting differences in site-specific prevalence, persistence, and bacterial load. These findings underscore the importance of longitudinal monitoring to inform targeted infection control strategies in LTCFs.

**Data availability statement:** Anonymized data that support the findings of this study are included as supporting information.

**Funding:** This work was supported by the Centers for Disease Control and Prevention (contract 75D30121C11191 to TP) and the National Institutes of Health (NIH) National Institute on Minority Health and Health Disparities (U54MD012388 to TP) and National Institute of Allergy and Infectious Diseases (R15AI156771 to TP). The funding agencies had no role in the study design, data collection, analysis, decision to publish, or preparation of the manuscript.

**Competing interests:** The authors have declared that no competing interests exist.

## Author summary

Nursing home acquired pneumonia is associated with a significant number of hospitalizations and deaths among long term care facility residents. While causes of bacterial pneumonia are well documented, limited research has addressed carriage of these pathogens over time. To understand these temporal patterns among long term care facility residents, we monitored nasal and oral carriage of *five respiratory pathogens (Haemophilus influenzae, Pseudomonas aeruginosa, Streptococcus pneumoniae, Staphylococcus aureus,* and *Chlamydia pneumoniae)* over two years. We found that positivity rates varied across time and between sample types for these pathogens, highlighting the need for continuous monitoring and pathogen specific approaches to guide infection control strategies in long term care facilities. These findings can help lower risks of nursing home acquired pneumonia and improve resident health outcomes.

## Introduction

Nursing home acquired pneumonia (NHAP) is a leading cause of morbidity and mortality in long term care facilities (LTCFs), accounting for 40% of hospitalizations and up to 33% of deaths among residents [1]. It is estimated that 365 out of every 1000 LTCF residents are diagnosed with pneumonia annually [2], a rate 6–11 times higher than the incidence of pneumonia in elderly people living in the community [3,4]. Further, the medical care and treatment required for NHAP contribute to significant healthcare costs due to the potential for prolonged hospital stays, need for intensive care, and treatment-resistant etiologies. While pneumonia can often be treated by LTCF staff at an average cost of approximately $3,700 [5], up to 30% of patients require hospitalization which increases average treatment cost to $10,400 [6,7]. For patients who require hospitalization, mortality rates range from 13-45% [8,9]. These high costs and poor outcomes underscore the need for targeted, evidence-based infection prevention and stewardship strategies in nursing homes [10].

Many cases of pneumonia in older adults are endogenous, caused by bacteria that asymptomatically colonize the upper respiratory tract [11–27]. These bacteria commonly include *Streptococcus pneumoniae, Staphylococcus aureus,* and *Haemophilus influenzae,* as well as *Pseudomonas aeruginosa,* which is not typically found in healthy individuals but may colonize those with compromised immune systems or chronic lung conditions, particularly in healthcare settings [28–36]. Autoinfection occurs when these colonizing pathogens are aspirated into the lower respiratory tract where they can initiate infection [37]. Aspiration can occur either as small-volume micro-aspiration (a common process even in healthy individuals, particularly during sleep) or as larger-volume macro-aspiration in individuals with impaired swallowing or altered consciousness [38]. Micro-aspiration is considered the primary mechanism underlying most cases of community and healthcare associated pneumonia, while macro-aspiration is a recognized cause of severe aspiration pneumonia in high-risk populations [39,40]. Older individuals are particularly vulnerable to both forms of

aspiration pneumonia due to age-related decline in humoral and cell-mediated immunity, impaired mucociliary clearance, increased likelihood of underlying airway disorders, and altered levels of consciousness [41–44]. These vulnerabilities are further compounded by co-morbidities (e.g., diabetes, COPD, dementia) and common medication use (e.g., sedatives, anticholinergics, proton pump inhibitors), which together diminish host defenses and increase the likelihood that bacteria will colonize and initiate infection. Given that many cases of pneumonia arise from aspiration of colonizing organisms, pre-ventions strategies should focus not only on reducing aspiration risk, but also on minimizing colonization with opportunistic pathogens.

Many pneumonia-causing bacteria colonize dental plaque biofilms [45–53], providing a potential source for micro- and macro-aspiration pneumonia [46,48]. Up to 80% correlation has been found between aerobic dental plaque colonization and the causative agent of aspiration associated pneumonia [48]. Unsurprisingly, poor oral hygiene increases the risk of pneumonia [46,54–56], highlighting its potential as an intervention target. Further, elderly LTCF residents tend to have par-ticularly poor oral health with high plaque scores [57], indicating that improving oral health among this population may be especially impactful. While promising, oral health interventions have shown mixed results in lowering NHAP incidence. In some studies, improved oral hygiene has been shown to decrease the incidence rate of pneumonia, potentially preventing as many as 10% of NHAP deaths [48,58–61]. Other studies have found no advantage in toothbrushing and chlorhexidine rinses over standard care in preventing NHAP [62]. Given that NHAP can be caused by a range of colonizing opportunistic pathogens, it is critical to characterize the specific carriage patterns and preferred colonization sites of each pathogen. Interventions targeting oral hygiene may only impact pathogens that colonize plaque or the oral cavity but will have limited impact on those that primarily colonize elsewhere. Understanding these pathogen-specific patterns is key to evaluating which interventions are likely to have the greatest impact.

To better understand the carriage patterns of colonizing opportunistic pathogens associated with NHAP, we conducted a longitudinal study of asymptomatic bacterial carriage in residents of long-term care facilities. Over a two-year period, we collected paired oral and nasal samples from 121 residents and characterized carriage of *S. pneumoniae, S. aureus, H. influenzae, and P. aeruginosa.* This study design allowed us to characterize not only prevalence over time, but also other colonization dynamics including persistence, recurrence, co-colonization, and factors associated with bacterial load. Unlike previous work that is largely cross-sectional and focused on hospitalized patients, our work provides insight into carriage patterns in a non-acute setting, laying the groundwork for developing targeted prevention and intervention strate-gies that reflect species-specific colonization behaviors.

## Methods

### Ethics statement

Approval for this project was granted by the Northern Arizona University Institutional Review Board (# 1766728). Partici-pant recruitment began Oct 27th of 2021 and continued until the last sampling event on Oct 31st of 2023. All participants provided formal written consent.

### Sample and data collection

Across three LTCFs in the Phoenix metropolitan area, 121 residents were enrolled to participate in this study. Participants provided their pneumonia vaccination status, as well as demographic information (age, sex, and ethnicity). Swabs of the anterior nares and oral mucosa were collected every other week from November 2021 to October 2023 (49 events total). Although the oral and nasal cavities may not serve as the primary reservoir for all the target pathogens, they have been previously found to colonize one or both sites [11,13,16,63–74]. We chose oral and nasal sampling over oropharyngeal or nasopharyngeal sampling, because they are less invasive and therefore better tolerated for frequent longitudinal sampling in elderly LTCF residents [75,76]. Oral hygiene professionals supervised the self-sampling, which has proven to be an

effective and sensitive method of detecting pathogens at these body sites [66]. Collected swabs were stored in 1mL of Liquid Amies at -20°C until processing.

## Multi-pathogen qPCR panel

Total DNA was extracted from nasal and oral swab samples using the Applied Biosystems MagMAX Viral and Pathogen Nucleic Acid Isolation Kit with the KingFisher Flex system, according to the manufacturer's instructions. qPCR was performed using a multi-pathogen panel (adapted from published assays [77–81]) designed to detect and quantify *H. influenzae*, *P. aeruginosa*, *S. pneumoniae*, *S. aureus,* and *Chlamydia pneumoniae* (S1 Table). We performed an in-silico PCR to evaluate the specificity of the primers and probes across a comprehensive set of reference genomes for each pathogen. To improve sensitivity and inclusivity, some oligonucleotide sequences were modified based on the in-silico results: degenerate bases were introduced at positions where common variants were observed among reference genomes, allowing broader detection while maintaining specificity (see S1 Table for final sequences and modifications). The assays were validated independently using positive controls and optimized to be run in duplex. There was predicted self-complementarity between oligos for *H. influenzae* and *S.* pneumoniae as well as *S. aureus* and *P. aeruginosa*, but no other combinations produced cross-primer dimerization based on in-silico PCR. Serial dilutions of quantitative genomic DNA were included in all runs to allow for relative DNA quantification using standard curve analysis.

Primers and probes were synthesized by Integrated DNA Technologies (Coralville, IA, USA). Each qPCR reaction was carried out in a 10 μL volume containing 5 μL of TaqMan Universal Master Mix, 2 μL of template DNA, final concentrations of 0.6 μM for primers and 0.3 μM for probes, and nuclease-free water. Cycling conditions were as follows: 50°C for 2 minutes, 95°C for 15 minutes, then 45 cycles of 95°C for 15 seconds and 57°C for 1 minute. Amplifications were completed using an Applied Biosystems QuantStudio 7 Pro System, and results were analyzed using the QuantStudio Design & Analysis Software (version 2.6.0). Non-template controls were included in qPCR reactions to ensure that contamination of reagents or equipment did not produce false positive results.

## Secondary validation using multi-pathogen amplicon sequencing

To confirm that the multiplex qPCR assays accurately identified the target species, we used a species-specific targeted amplicon sequencing approach to detect additional genomic regions unique to each pathogen in a subset of qPCR positive samples (S2 Table). The multiplex PCR was performed using the KAPA 2G Fast Multiplex PCR Master Mix in 25μL reaction volumes, with 5ng of template DNA, 0.2 μM concentration of each primer, and water to reach the final volume. PCR thermocycling conditions consisted of an initial denaturation at 95°C for 3 minutes, followed by 35 cycles of denaturation at 95°C for 15 seconds, annealing at 60°C for 30 seconds, and extension at 72°C for 1 minute and 30 seconds and a final extension at 72°C for 1 minute. PCR products were prepared for sequencing using our standard amplicon sequencing protocol [82] and were sequenced on the Illumina MiniSeq platform. Reads were aligned to reference genomes (Accessions in S2 Table) using BWA-MEM v0.7.17 [83].

## Data analysis

All data cleaning and statistical analyses were conducted using R version 4.4.1. Statistical significance was set at a p-value of less than 0.05 for all tests. To minimize bias from sampling inconsistency, results are only included from participants who attended at least 1/3 of sampling events.

McNemar's test [84] was used to compare colonization rates between nasal and oral samples for each pathogen, accounting for the paired nature of the samples collected from the same participants. Matched odds ratios with 95% confidence intervals were calculated to quantify the magnitude of differences in colonization rates. To estimate the probabilities of transitioning between colonization states over time, we employed a Markov chain analysis with bootstrapping

(1000 for each pathogen). The resulting transition matrices generated from bootstrapping were averaged to produce a final transition matrix representing the mean transition probabilities. Linear mixed models were fitted using the lme4 [85] and lmerTest [86] packages to quantify predictors of average relative DNA quantities. Full models included transient vs. persistent colonization, age, sex, pneumococcal vaccination, and recent exposure (estimated as the proportion of positive participants from the same LTCF during the previous sampling event) as fixed effects as well as random intercepts for each participant to account for repeated measures. Participants who were never colonized by the pathogen of interest were excluded from this analysis to focus on the comparison between transient and persistent colonization. The results are reported in relative units (RQ), indicating fold changes in bacterial abundance relative to a standardized control.

To assess co-colonization, we identified instances where one or more pathogens were detected simultaneously for an individual. For each pathogen combination, we calculated the observed frequency of co-colonization. To evaluate whether these rates were greater than expected by chance, we computed expected co-colonization rates under the assumption of independence using the product of each pathogen's marginal positivity probability and the probability of being positive for at least one of the other pathogens. We also conducted pairwise comparisons to determine whether specific pathogen pairs were detected together more frequently than expected under independence. To quantify uncertainty and test for statistical significance, we used a bootstrap resampling approach (10,000 iterations) to generate 95% confidence intervals and we compared observed versus expected co-colonization rates using a two-tailed test. Co-colonization patterns were visualized with an upset plot created with the UpSetR [87] package, which visualizes the frequency of each pathogen combination. To investigate potential temporal associations between colonization with different pathogens, we performed a correlation analysis incorporating current and lagged colonization statuses across timepoints.

## Results

### Participant description

The 121 participants ranged in age from 60 to 97 (median = 83, mean = 82.97) and were a majority female (76.03%). Race/ethnicity was not included in this analysis due to a significantly imbalanced distribution within the sample; approximately 96% of participants were non-Hispanic whites. The subset of participants who attended at least one-third of sampling events (n = 85) closely matched the demographic distribution of the overall study population, with a mean age of 83 and approximately 3/4 being female. These individuals attended an average of 34 out of 49 sampling events, approximately 70% were vaccinated against streptococcal pneumonia, and were evenly distributed between the three LTCFs. Deidentified participant metadata, including age, sex, vaccination status and LTCF site number are provided in S3 Table.

### Colonization rates at nasal and oral sites

We observed wide ranges of colonization rates for the pathogens of interest and significant site-specificity towards the nasal or oral cavity. Of the approximately 6,000 samples collected from the 85 participants in LTCFs, 9.99% of samples were positive for *H. influenzae*, 5.78% were positive for *P. aeruginosa*, 22.07% were positive for *S. pneumoniae*, and 18.01% were positive for *S. aureus*. Notably, *C. pneumoniae* was not detected in any samples. Table 1 provides an overview of the colonization metrics for each pathogen. Across time points, notable variation was observed in both

**Table 1. Colonization metrics by pathogen.**

| Pathogen | Average weekly sample positivity (range) | Average weekly individual positivity (range) | Cumulative incidence |
|---|---|---|---|
| *H. influenzae* | 9.81% (0–23.48) | 17.75% (0–39.39) | 81.18% |
| *P. aeruginosa* | 5.56% (0–20) | 9.95% (0–37.74) | 83.53% |
| *S. pneumoniae* | 21.70% (0.89–31.62) | 31.89% (1.79–41.67) | 78.82% |
| *S. aureus* | 17.80% (0–36.13) | 28.00% (0-55.36) | 91.76% |

sample-level and individual-level positivity rates. The cumulative incidence values illustrate that a majority of participants were colonized by at least one of these pathogens during the study period.

Our findings indicate that these pathogens differentially colonize the oral and nasal cavities, with *H. influenzae* and *S. pneumoniae* predominating in the oral cavity, and *P. aeruginosa* and *S. aureus* more frequently detected in the nasal cavity (Table 2). These differences were statistically significant (McNemar's test, $p < 1 \times 10^{-6}$), suggesting that the observed distribution patterns are not due to random variation but reflect true biological differences in site-specific colonization. Matched odds ratios further quantify these disparities, demonstrating substantially increased odds of oral colonization for *H. influenzae* and *S. pneumoniae*, and notably increased odds of nasal colonization for *P. aeruginosa* and *S. aureus*.

## Longitudinal colonization rates

There were statistically significant changes in participant positivity over time for some of the pathogens (Fig 1). *H. influenzae* colonization decreased over time in both sample types, *S. aureus* colonization decreased in nasal samples over time, and *P. aeruginosa* and *S. pneumoniae* colonization increased in nasal samples over time ($p < 0.01$). Oral colonization with *P. aeruginosa, S. pneumoniae,* and *S. aureus* did not significantly change over time. Colonization at the individual level changed significantly over time for *H. influenzae, P. aeruginosa,* and *S. aureus* ($p < 0.01$). These changes were not correlated with any measured variables and there were no changes in protocol during the study. The longitudinal colonization data for each participant is provided in S4 Table.

**Table 2. Site-specific colonization.**

| Pathogen | Positive Oral Samples | Positive Nasal Samples | Odds Ratio (95% CI) | Interpretation |
|---|---|---|---|---|
| *H. influenzae* | 16.67% | 3.35% | 9.81 (7.14–13.47) | 881% higher odds in oral samples |
| *S. pneumoniae* | 30.78% | 13.50% | 10.23 (7.70–13.60) | 923% higher odds in oral samples |
| *P. aeruginosa* | 4.28% | 7.31% | 1.98 (1.53–2.56) | 98% higher odds in nasal samples |
| *S. aureus* | 11.41% | 25.05% | 4.75 (3.83–5.89) | 375% higher odds in nasal samples |

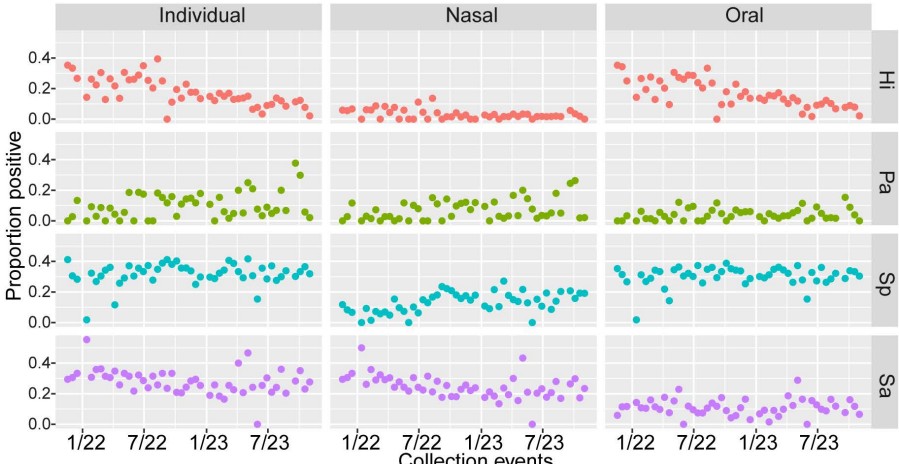

**Fig 1. The proportion of individuals (from oral, nasal, or both sample types), nasal samples, and oral samples testing positive for each pathogen over the course of the study.** Pathogens include *H. influenzae* (Hi, red), *P. aeruginosa* (Pa, green), *S. pneumoniae* (Sp, blue), and *S. aureus* (Sa, purple). Each dot represents the positivity rate for a collection event, with trends highlighting temporal dynamics in colonization prevalence across anatomical sites and pathogens.

## Duration of colonization

The distribution of time participants spent colonized shows notable variability among pathogens (Fig 2). The proportion of time that participants spent colonized by *S. pneumoniae* and *S. aureus* exhibited a pronounced bimodal pattern with a peak near zero, indicating never or highly transient colonization, and another near one, reflecting persistent colonization. Interestingly, colonization by *P. aeruginosa* appears to be predominantly short-lived, with only rare cases of persistent colonization or carriage. Colonization with *H. influenzae* was similarly skewed, though less drastically. The observation that individuals tend to be colonized by *S. aureus* either a majority of the time or only transiently has previously been found [88,89]. Less is known about persistence of colonization with the other pathogens of interest in healthy individuals. To quantify this variability, participants were categorized as "never" (no colonization over the study), "transient" (colonization in <50% of sampling events), or "persistent" (colonization in ≥50% of sampling events). For *H. influenzae*, 67.1% of participants experienced transient colonization, 18.8% were never colonized, and 14.1% were persistently colonized, indicating that *H. influenzae* colonization is mostly short-lived for otherwise healthy individuals. For *P. aeruginosa*, 82.4% were transient, 16.5% were never colonized, and 1.18% were persistently colonized, indicating that *P. aeruginosa* is not typically a long-term colonizer. In contrast, *S. pneumoniae* exhibited a more balanced pattern: 43.5% were transiently colonized, 21.2% were never colonized, and 35.3% were persistently colonized. For *S. aureus*, only 8.24% were never colonized, with 71.8% transiently colonized and 20% persistent colonization.

The average duration of a colonization episode — defined as a continuous stretch of consecutive positive detections at either of the two body sites — was approximately 4 weeks for *H. influenzae*, 2.4 weeks for *P. aeruginosa*, 9.6 weeks for *S. pneumoniae*, and 8.8 weeks for *S. aureus* (among individuals who were colonized at least once during the study). Recurrent colonization (recolonization after pathogen was not detected and thus assumed to be cleared from an individual) was observed in 58.8% of participants for *H. influenzae*, 61.2% for *P. aeruginosa*, 57.6% for *S. pneumoniae*, and 72.9% for *S. aureus*. These colonization patterns are illustrated below (Fig 3). In our population, *S. pneumoniae* and *S. aureus* are often persistent colonizers, while *H. influenzae* and *P. aeruginosa* typically appear transiently. Interestingly, persistent colonization with *S. aureus* tends to be in the nasal cavity while persistent colonization with *S. pneumoniae* tends to occur in the oral cavity (Fig 4).

Markov chains were used to provide a more nuanced understanding of colonization likelihood and stability over time. For *H. influenzae*, at any given time point, the probability of becoming colonized was 9.05%, with a 55.92% chance of

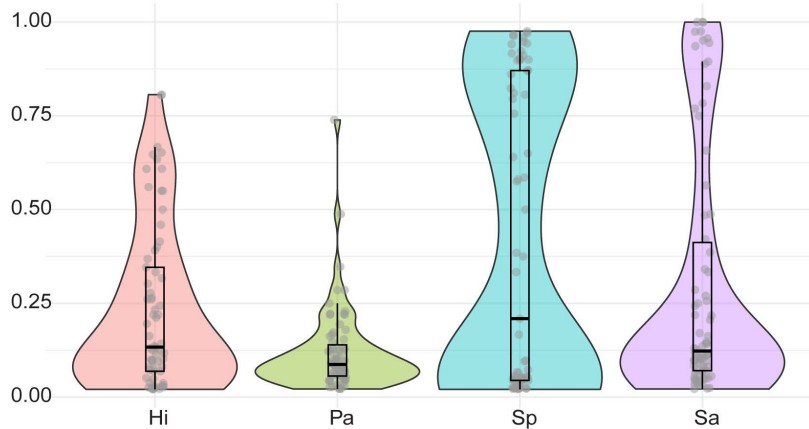

**Fig 2. The proportion of events participants spent colonized by each pathogen (if colonized at least once); *H. influenzae* (Hi, red), *P. aeruginosa* (Pa, green), *S. pneumoniae* (Sp, blue), and *S. aureus* (Sa, purple).** The distribution indicates variability in colonization duration across pathogens, with box plots embedded to show median and quartile ranges.

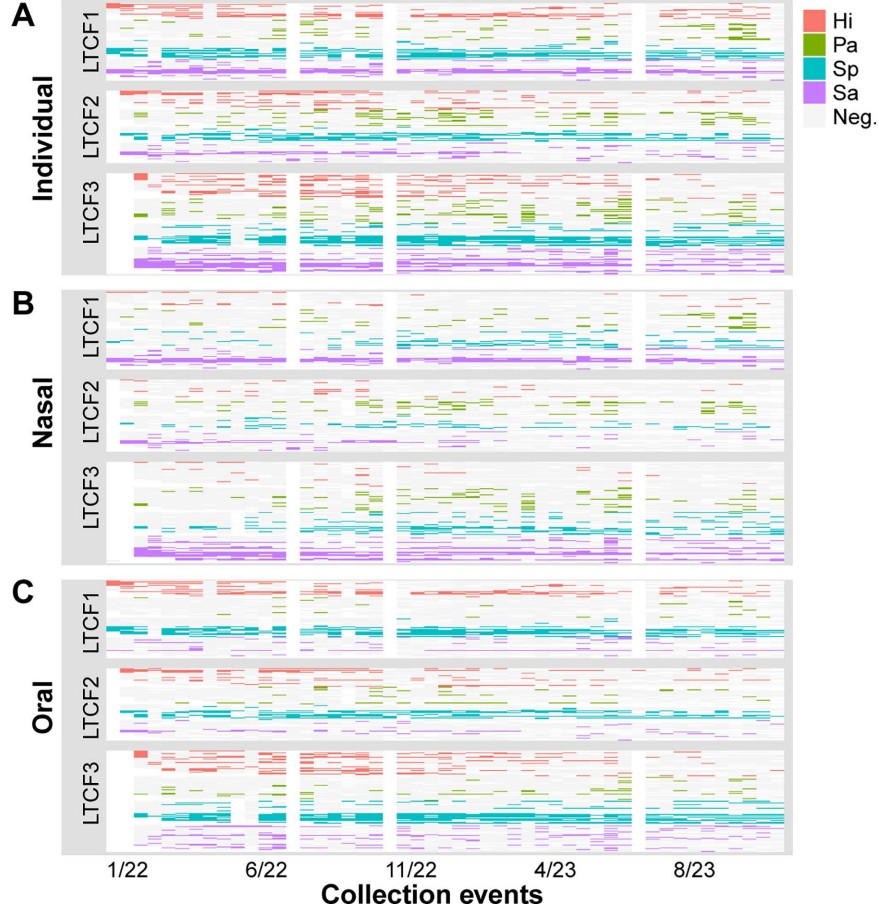

**Fig 3. Intra-facility colonization patterns of *H. influenzae* (Hi, red), *P. aeruginosa* (Pa, green), *S. pneumoniae* (Sp, blue), and *S. aureus* (Sa, purple) across the study period. A)** Individual-level colonization patterns (based on detection in oral, nasal, or both sample types), **B)** Nasal colonization, and **C)** Oral colonization, with each row representing a participant grouped by LTCF. The figure illustrates the variability in pathogen persistence over time and clustering of colonization events within the facilities.

maintaining colonization once established, indicating moderate persistence. *P. aeruginosa* showed a similar colonization probability of 9.05%, but only a 21.67% probability of remaining colonized, reflecting its low likelihood of sustained colonization. In contrast, *S. pneumoniae* had a 7.56% probability of colonizing but an 83.64% chance of remaining colonized, suggesting that *S. pneumoniae* is more persistent once established. For *S. aureus*, the colonization probability was 10.74%, with a 72.98% likelihood of continuing colonization once established.

### Bacterial load

To identify which factors most influenced bacterial load (and thus the amount of bacterial material available for aspiration), relative DNA quantities during positive weeks were analyzed using linear mixed models for each pathogen. For all pathogens except *P. aeruginosa*, a subject's categorization as a transient or persistent carrier was the most significant predictor of bacterial load, with persistent carriers showing higher bacterial levels (Table 3). Given that there was not a pronounced group of persistent *P. aeruginosa* carriers, the absence of a significant relationship aligns with expectations. For both *H. influenzae* and *P. aeruginosa,* bacterial load was also significantly associated with recent high levels of exposure within

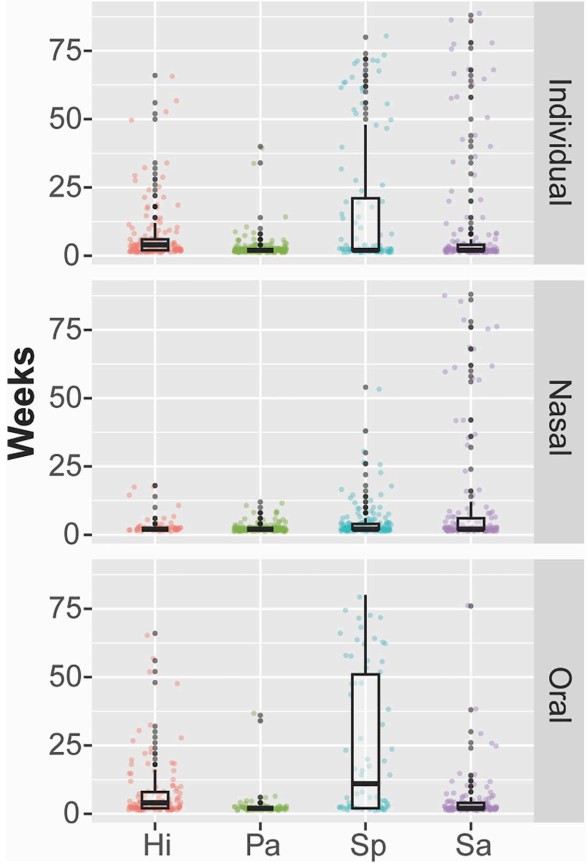

**Fig 4. Duration of continuous colonization for each pathogen—H. influenzae (Hi, red), P. aeruginosa (Pa, green), S. pneumoniae (Sp, blue), and S. aureus (Sa, purple)—at the individual level and at nasal and oral sites, highlighting variability.**

**Table 3. Predictors of bacterial load. Summary of linear mixed-effects model results assessing the relationship between bacterial load and predictor variables including age, sex, pneumococcal vaccination status, carriage type (transient vs persistent), and exposure to other positive individuals in the same LTCF. Separate models were run for each pathogen. Only statistically significant variables (p<0.05) are shown.**

| Pathogen | Predictor | Estimate | Std. Error | T value | P-value | Significance |
|---|---|---|---|---|---|---|
| *H. influenzae* | Transient carriage | -364.2 | 79.2 | -4.60 | **< 0.001** | *** |
| *H. influenzae* | Exposure in LTCF | 651.9 | 159.1 | 4.10 | **< 0.001** | *** |
| *P. aeruginosa* | Exposure in LTCF | 1975.5 | 980.4 | 2.02 | **0.044** | * |
| *S. pneumoniae* | Transient carriage | -898.7 | 302.8 | -2.97 | **0.005** | ** |
| *S. aureus* | Transient carriage | -232.2 | 15.2 | -15.24 | **< 0.001** | *** |

the LTCF based on facility-wide positivity rates two weeks prior. In our study, these two organisms exhibited less stable colonization compared to *S. pneumoniae* and *S. aureus*, suggesting that their bacterial load may be more strongly influenced by recent transmission or environmental influxes.

## Co-colonization

Co-colonization was observed in nearly one-third of samples that tested positive for at least one of the four pathogens. The most frequent combinations involved *S. pneumoniae* co-detected with either *H. influenzae* or *S. aureus,* followed

by combinations involving *P. aeruginosa* with either *H. influenzae, S. pneumoniae,* or *S. aureus,* and *S. aureus* with *H. influenzae* (Fig 5). Given the high carriage rates of some pathogens, many co-colonization events are expected to occur purely by chance; therefore, we tested whether the observed rates differed from what would be expected under statistical independence. We found no significant difference in the co-colonization rates (p > 0.4) and the pairwise comparisons between pathogens showed no statistically significant enrichment in co-detection frequencies (p > 0.4). Lagged colonization analysis also revealed no strong associations between pathogens across time points (r < 0.2), suggesting that colonization by one pathogen does not increase or decrease the probability of colonization by another pathogen.

## Discussion

Findings from this study highlight several important colonization patterns among LTCF residents, all of which underscore the value of longitudinal data in revealing temporal variability and individual-level dynamics that would be impossible to capture with cross-sectional approaches. This study is strengthened by sample collection from two body sites (oral and nasal), investigation of multiple pathogens, and the utilization of qPCR for pathogen detection, which has been shown to be an especially reliable method of detection in older populations [90]. Longitudinal data collection provides valuable information on how prevalence, persistence, bacterial load, and co-colonization patterns are affected temporally and on an individual and institutional basis. Additionally, this study provides valuable information about asymptomatic carriage in otherwise healthy individuals; the very few existing longitudinal NHAP research studies have been limited to hospitalized patients. These insights can guide the design of effective mitigation strategies for NHAP and associated etiological agents.

In this study, we characterized the carriage patterns of four colonizing opportunistic pathogens associated with NHAP. *S. pneumoniae* and *S. aureus* are widely reported as colonizers of the upper respiratory tract [13,91–94]; while *H. influenzae* colonization is detected less frequently and considered more transient in carriage [72,95–97]. *P. aeruginosa* is generally considered an environmental organism that is associated with disease in immunocompromised individuals or those with structure lung abnormalities (e.g., COPD or bronchiectasis) [74,98–100]. Carriage rates in our cohort varied considerably across the four pathogens, with *S. pneumoniae* and *S. aureus* exhibiting the highest average weekly positivity (31.89% and 28.00%, respectively), followed by *H. influenzae* (17.75%) and *P. aeruginosa* (9.95%). These rates exceeded many prior cross-sectional estimates in older adults, which typically report *S. pneumoniae* carriage rates under 10% (although higher rates have also been reported) [23,93,96,101–103], *S. aureus* at 20–35% [12,104,105], *H.*

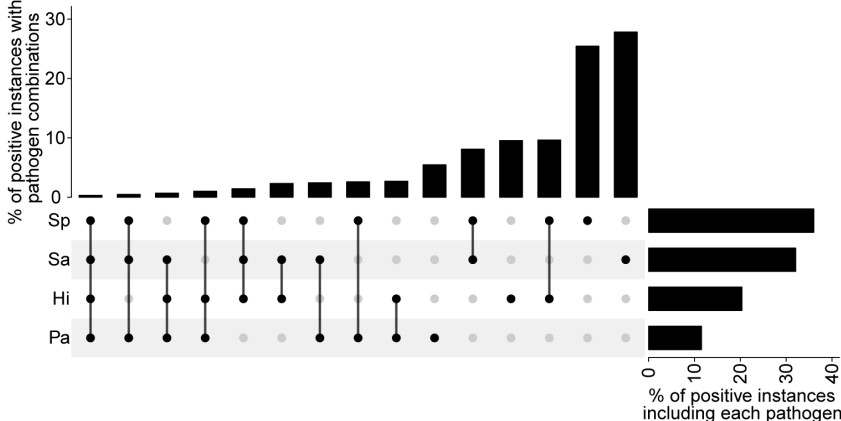

**Fig 5. Co-colonization observed throughout the study within individuals (i.e., considering both oral and nasal results).** The bar chart at top shows the percent of instances where individuals who were positive for at least one pathogen were positive for a given combination of pathogens (filled black dots). Each row represents a pathogen, and the bar chart to the right shows the relative prevalence for each pathogen among individuals that tested positive for at least one pathogen.

*influenzae* around 2–10% [72,96], and *P. aeruginosa* at 2–7% in healthy populations [36], though higher rates have been observed in hospitalized older adults (up to 43%) and healthcare workers [106–108]. The higher carriage rates in our study may reflect increased detection sensitivity afforded by molecular detection methods and the sampling of more than one body site. Our longitudinal approach also revealed substantial week-to-week variation in detection rates (e.g., *S. pneumoniae* ranging from 0.89% to 31.62% and *S. aureus* from 0% to 36.13%), highlighting the limitations of using single cross-sectional samples to estimate carriage rates. In addition, we observed significant changes in positivity over time for several pathogens despite no change in study methodology or any other measured variables. These findings emphasize that point-prevalence measures alone underestimate the true burden of colonization, potentially leading to an incomplete understanding of pathogen carriage.

The relatively high detection rate of *P. aeruginosa* warrants careful interpretation, as this organism is not a typical colonizer of the upper respiratory tract in healthy humans and is ubiquitous in the environment. Although several participants exhibited repeated positive detections over multiple weeks — suggesting true establishment of colonization in some cases — many detections occurred at only a single time point. Although we included non-template controls throughout extraction and amplification steps to rule out technical contamination, the sporadic carriage pattern raises the possibility that some detections may reflect environmental contamination or short-lived exposure rather than truly established colonization. Further investigation is needed to determine whether *P. aeruginosa* can consistently establish colonization in this population or whether its detection in upper respiratory samples reflects exposure to environmental sources.

All four pathogens examined in this study were detected in both the nasal and oral cavities; however notable site-specific trends emerged. Consistent with prior studies, *S. aureus* was more commonly detected in the nasal cavity, though oral carriage was also frequent [13,66,109–111]. *P. aeruginosa* was also more common in the nasal cavity, and although upper respiratory colonization is less well characterized in healthy individuals, prior studies have also reported its presence in nasopharyngeal, oropharyngeal, and denture biofilms [36,99,112]. In contrast, *S. pneumoniae* and *H. influenzae* were more frequently detected in oral samples. While the nasopharynx is traditionally considered the primary reservoir for *S. pneumoniae* and *H. influenzae,* they have been detected in oral and oropharyngeal samples, sometimes with at higher rates in older adults [23,58,63,95,101,113–116]. These findings reinforce the importance of sampling multiple anatomical sites to more comprehensively characterize carriage patterns and understand site-specific colonization preferences.

Carriage duration varied across each pathogen and showed distinct patterns within each pathogen. In our cohort, the average duration of a colonization episode was longest for *S. pneumoniae* (9.6 weeks), followed by *S. aureus* (8.8 weeks), *H. influenzae* (4 weeks) and *P. aeruginosa* (2.4 weeks). Both *S. aureus* and *S. pneumoniae* exhibited distinct groups of persistent carriers (20% and 35.3%, respectively) and transient carriers (71.8% and 21.2%, respectively). Other reports have estimated that 20% of adults are intermittent carriers of *S. pneumoniae* and 10% are persistent carries, with a median duration of 7 weeks [117]. *S. aureus* carriage is also known to vary, with 20–30% of individuals showing persistent nasal carriage for months to years and 30–50% showing transient carriage patterns [13,89,91]. In contrast, less is known about the duration of *H. influenzae* and *P. aeruginosa* carriage in healthy adults, but both are generally considered to be less frequent colonizers [36,72,96]. Our findings support this with persistent colonization being relatively rare for *P. aeruginosa* (1.18% vs. 82.4% transient) and *H. influenzae* (14.1% vs 67.1% transient) and both exhibiting shorter colonization episodes and lower probability of continued carriage once established.

Bacterial load varied across pathogens and was influenced by distinct factors. For the more stable colonizers, *S. pneumoniae* and *S. aureus,* persistent carriers exhibited significantly higher bacterial loads than transient carriers, suggesting that persistent carriage is associated with more successful establishment of these species and maintenance of higher density in the established niche. In contrast, for the less stable colonizers, *H. influenzae* and *P. aeruginosa,* recent exposure within the facility was a strong predictor of bacterial load. This distinction aligns with their broader colonization patterns as both *H. influenzae* and *P. aeruginosa* exhibited significant variation in positivity rates over time, suggesting that they may be more sensitive to fluctuations in environmental or interpersonal exposure. These findings support the

idea that *S. pneumoniae* and *S. aureus* tend to establish more stable, self-sustaining carriage, while *H. influenzae* and *P. aeruginosa* colonization may be driven more by exposure.

Co-colonization by multiple pathogens has important implications for pneumonia risk, as polymicrobial infections may complicate disease progression, increase severity, and reduce treatment efficacy. Polymicrobial pneumonia is reported in up to 40% of cases with an identified cause [118], raising the question of whether co-colonization in the upper respiratory tract is an important risk factor. Previous studies have reported positive associations between *S. pneumoniae* and *H. influenzae* colonization, and negative associations between *S. aureus* with *S. pneumoniae* and *H. influenzae* [119,120]. In our study, co-colonization occurred in approximately 1/3 of colonization events, suggesting that it is relatively common. However, the observed rates did not significantly differ from what would be expected by random chance assuming independence between the pathogens. We found no evidence that colonization by one pathogen increased or decreased the likelihood of subsequent colonization by another. Together, these findings indicate that while co-colonization occurred frequently, there was no evidence that this was driven by facilitative or antagonistic microbial interactions.

In addition to the four primary pathogens examined in this study, we tested all samples for *C. pneumoniae,* but it was not detected in any samples across the two-year study. *C. pneumoniae* is a common cause of upper and lower respiratory tract infections and is frequently implicated in community-acquired pneumonia [121]. While many infections are thought to be asymptomatic, chronic infections or colonization of the upper airway is not commonly reported in healthy adults though it has been linked to other diseases [122]. *C. pneumoniae* has been detected in throat swabs or sputum during acute illness or active infection, but detection in healthy, asymptomatic individuals is rare [122,123], and most prevalence estimates are based on serologic evidence of past exposure rather than direct sampling of the respiratory tract [124–126]. Moreover, as an obligate intracellular pathogen, *C. pneumoniae* is less likely to be recovered via surface swabbing methods optimized for extracellular organisms [126]. Its absence in both nasal and oral swabs from our asymptomatic LTCF cohort is therefore consistent with previous findings and may reflect both the low carriage rate or exposure in this population and methodological imitations in detection.

This study has several limitations that should be considered when interpreting the findings. First, the study was conducted in LTCFs within a single geographic region (the Phoenix metropolitan area in Arizona), which may limit the generalizability of the results to other regions or care settings with different population demographics or care practices. Second, only two anatomical sites (oral and nasal cavities) were sampled; while these sites were chosen to balance pathogen detection with participant comfort during frequent longitudinal sampling, additional sites (e.g., oropharynx, nasopharynx) would provide a more comprehensive understanding of the primary reservoirs and colonization patterns for these pathogens. This may have introduced a sampling bias and potentially underestimated the overall burden of colonization as studies have shown that increasing the number of body sites sampled increases prevalence estimates [66]. Third, we did not perform strain-level genotyping or whole-genome sequencing, which limits our ability to assess intra-species diversity and strain-specific colonization patterns. It is likely that different strains may exhibit distinct site-specificity and colonization patterns, and this is an important area for future investigation. Finally, we did not collect detailed data on participant co-morbidities or medication use, which are known to influence mucosal immunity and the likelihood of pathogen colonization in older adults. These limitations highlight important areas for future work, including expansion to LTCFs in more regions, broader anatomical sampling, integration of strain-level analysis, and collection of host-level clinical data to better understand the complex interactions between host factors and pathogen carriage.

This study highlights the complex dynamics of colonization in LTCFs, including variability in site-specific colonization, the distinction between transient and persistent colonization, and the occurrence of co-colonization. Longitudinal data collection was crucial, enabling a better understanding of colonization dynamics and persistence. These insights emphasize the importance of targeted surveillance and intervention strategies that consider the temporal and specific nature of colonization, particularly for high-risk pathogens like *S. aureus* and *P. aeruginosa*.

## Supporting information

**S1 Table. Multi-pathogen qPCR assay.** Previously published primers and probes for detecting *H. influenzae, C. pneumoniae, P aeruginosa, S. pneumoniae,* and *S. aureus* were used in this assay, with slight modifications to sequences to capture variation present in published reference sequences. In silico PCR was performed against all complete genome sequences available in NCBI's RefSeq database; the number of reference sequences for each species indicated in the Notes column. In cases where a substantial proportion of the reference genomes contained a base mismatch at a primer or probe site, a degenerate base was added to the primer sequence to improve inclusivity across variants.
(XLSX)

**S2 Table. Primers used for targeted amplicon sequencing.** The genomic targets included established MLST targets as well as additional pathogen-specific targets identified by selecting conserved regions from reference genome alignments within each species. There was a total of 106 primer pairs: 11 for *C. pneumoniae,* 9 for *H. influenzae,* 36 for *P. aeruginosa,* 30 for *S. aureus,* and 20 for *S. pneumoniae.*
(XLSX)

**S3 Table. Deidentified demographic and vaccination data for study participants.** Table includes participant age, sex, LTCF site number, and pneumococcal vaccine status.
(XLSX)

**S4 Table. Longitudinal colonization data for each participant.** Table includes binary and quantitative qPCR-based colonization data for *H. influenzae, P. aeruginosa, S. pneumoniae,* and *S. aureus* across all sampling events.
(XLSX)

## Author contributions

**Conceptualization:** Ryann N. Whealy, Tara N. Furstenau, Talima Pearson, Viacheslav Y. Fofanov.

**Data curation:** Alexander Roberts, Tara N. Furstenau, Skylar Timm, Sara Maltinsky, Sydney J. Wells, Kylie Drake, Kayla Ramirez.

**Formal analysis:** Ryann N. Whealy, Tara N. Furstenau.

**Funding acquisition:** Talima Pearson, Viacheslav Y. Fofanov.

**Investigation:** Alexander Roberts, Skylar Timm, Sara Maltinsky, Sydney J. Wells, Kylie Drake, Kayla Ramirez.

**Methodology:** Ryann N. Whealy, Tara N. Furstenau, Viacheslav Y. Fofanov.

**Project administration:** Ryann N. Whealy, Sara Maltinsky, Candice Bolduc, Ann Ross.

**Resources:** Tara N. Furstenau, Candice Bolduc, Ann Ross, Talima Pearson, Viacheslav Y. Fofanov.

**Supervision:** Talima Pearson, Viacheslav Y. Fofanov.

**Validation:** Alexander Roberts.

**Visualization:** Ryann N. Whealy, Tara N. Furstenau.

**Writing – original draft:** Ryann N. Whealy, Alexander Roberts.

**Writing – review & editing:** Ryann N. Whealy, Alexander Roberts, Tara N. Furstenau, Talima Pearson, Viacheslav Y. Fofanov.

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
