## [Decision Letter · Decision Letter 0]

22 May 2025

PGPH-D-25-00433

Longitudinal prevalence and co-carriage of pathogens associated with nursing home acquired pneumonia in three long-term care facilities

Dear Dr. Whealy,

Thank you for submitting your manuscript to PLOS Global Public Health. After careful consideration, we feel that it has merit but does not fully meet PLOS Global Public Health’s publication criteria as it currently stands. Therefore, we invite you to submit a revised version of the manuscript that addresses the points raised during the review process.

Thank you for submitting your manuscript to PLOS Global Public Health. The reviewers commend the study’s robust longitudinal design, clinical relevance, and methodological rigour. We are pleased to invite you to submit a revised version addressing the reviewers’ suggestions. These include clarifying methodological details, enhancing data presentation, and expanding the discussion of pathogen-specific dynamics and limitations.

We look forward to receiving your revised manuscript.

Kind regards,

Ben Pascoe

Academic Editor

Journal Requirements:

Additional Editor Comments (if provided):

Reviewers' comments:

Reviewer's Responses to Questions

**Comments to the Author**

1. Does this manuscript meet PLOS Global Public Health’s publication criteria?

Reviewer #1: Yes

Reviewer #2: Yes

Reviewer #3: Yes

2. Has the statistical analysis been performed appropriately and rigorously?

Reviewer #1: Yes

Reviewer #2: Yes

Reviewer #3: Yes

3. Have the authors made all data underlying the findings in their manuscript fully available (please refer to the Data Availability Statement at the start of the manuscript PDF file)?

Reviewer #1: Yes

Reviewer #2: Yes

Reviewer #3: Yes

4. Is the manuscript presented in an intelligible fashion and written in standard English?

Reviewer #1: Yes

Reviewer #2: Yes

Reviewer #3: Yes

Reviewer #1: This study examines the longitudinal colonisation dynamics of key respiratory pathogens among residents of three long-term care facilities in Phoenix, Arizona. Over two years, nasal and oral swabs from 121 participants were collected biweekly and analysed by qPCR (and supplemental amplicon sequencing) to quantify the prevalence, persistence, bacterial load, and co-colonisation of Haemophilus influenzae, Pseudomonas aeruginosa, Streptococcus pneumoniae, and Staphylococcus aureus. The findings reveal significant site-specific colonisation patterns and differences in persistence, with implications for targeted infection control strategies in nursing homes.

Strengths:

• Longitudinal Design: Two-year follow-up with frequent sampling provides robust temporal data.

• Methodological Rigor: Use of qPCR and confirmatory amplicon sequencing strengthens pathogen detection.

• Multiple Anatomical Sites: Sampling from both nasal and oral cavities highlights site-specific colonisation dynamics.

• Detailed Statistical Analysis: Employment of mixed models and Markov chain analyses enhances interpretation of persistence and transition probabilities.

• Clinical Relevance: Findings are directly applicable to improving infection control strategies in LTCFs.

Weaknesses:

• Generalisability: Study is limited to LTCFs in one geographic region, potentially affecting broader applicability.

• Sampling Scope: Only two body sites were sampled, which might underestimate overall colonisation.

• Pathogen Resolution: Lack of strain-level genotyping limits understanding of intra-species variability.

• Detection Limitations: The absence of Chlamydia pneumoniae detection is noted but not fully explored.

Suggestions for Improvement:

• Expand Geographic Scope: Consider including additional LTCFs from diverse regions to enhance generalisability.

• Broaden Sampling: Incorporate additional anatomical sites (e.g., throat, skin) to capture a more complete picture of colonisation.

• Strain-Level Analysis: Integrate genotyping or whole-genome sequencing to distinguish strain-specific dynamics.

• Address Negative Findings: Provide further discussion or additional experiments to clarify the non-detection of Chlamydia pneumoniae.

• Clarify Limitations: Expand on potential sampling biases and confounding factors and discuss how these might influence the results.

Reviewer #2: In, “Longitudinal prevalence and co-carriage of pathogens associated with nursing home acquired pneumonia in three long-term care facilities,” the main aim of the study was to understand colonization dynamics of respiratory pathogens in LTCF residents. The manuscript read with good flow and had all important information. It provided the rational of performing the research and the impact for longitudinal monitoring for informed targeted control strategies in this demographic population of long-term care residents.

Only a few considerations:

- Methods

o Line [120] [-160]: consider putting the manufacturer’s address of the assays used from them.

o Line [138-140]: Potentially mention briefly how there were adjustments made for the oligonucleotide sequences or provide references that do.

- Figures

-- Consider making figure 3 a bit bigger to see the left-side figure.

Reviewer #3: The study by Ryann N. Whealy et al. investigates the colonization dynamics of five respiratory pathogens among residents in three long-term care facilities (LTCFs) in Phoenix, Arizona, to better understand risks associated with nursing home acquired pneumonia (NHAP). Over a two-year period, oral and nasal swabs were collected and tested using qPCR for Haemophilus influenzae, Pseudomonas aeruginosa, Streptococcus pneumoniae, Staphylococcus aureus, and Chlamydia pneumoniae. The study found varying prevalence and persistence rates among the pathogens, with S. pneumoniae and S. aureus being more persistent and associated with higher bacterial loads. Pathogen colonization was site-specific, with some preferring the oral cavity and others the nasal cavity, and co-colonization occurred in approximately 20% of cases. These findings emphasize the value of longitudinal surveillance to inform targeted infection control strategies in LTCFs and potentially reduce NHAP incidence.

I have several suggestions to improve the clarity of the manuscript, as outlined below:

Results section:

Lines 253–259: For the results regarding the factors most strongly influencing bacterial load, please include a figure or table to clearly illustrate the data.

Discussion section:

Lines 293–295: Were there any changes in sampling methods or study protocols over the two-year observation period?

Additionally, please discuss the differences between persistent and transient colonizers, as well as the specific niches each pathogen predominantly colonized. Consider addressing how the biological characteristics of these bacteria may explain their colonization patterns.

**Do you want your identity to be public for this peer review?** For information about this choice, including consent withdrawal, please see our Privacy Policy

Reviewer #1: No

Reviewer #2: **Yes: ** Madison Goforth

Reviewer #3: No

---

## [Editor Report · Decision Letter 1]

9 Jul 2025

Longitudinal prevalence and co-carriage of pathogens associated with nursing home acquired pneumonia in three long-term care facilities

PGPH-D-25-00433R1

Dear Prof. Furstenau,

We are pleased to inform you that your manuscript 'Longitudinal prevalence and co-carriage of pathogens associated with nursing home acquired pneumonia in three long-term care facilities' has been provisionally accepted for publication in PLOS Global Public Health.

Best regards,

Ben Pascoe

Academic Editor

The authors have addressed all reviewer comments comprehensively. They’ve made appropriate revisions to the manuscript, including clarification of methods, additional contextual discussion, improved figures and tables, and a more detailed account of limitations and colonisation dynamics.